# Arabinoxylan-Based Bioplastic from Wheat Bran: A Promising Replacement for Synthetic Plastics

**DOI:** 10.3390/polym17182488

**Published:** 2025-09-15

**Authors:** Md Abdur Rahim Badsha, Michael Kjelland, Chad Ulven, Khwaja Hossain

**Affiliations:** 1Division of Science and Mathematics, Mayville State University, Mayville, ND 58257, USA; mdabdurrahim.badsha@mayvillestate.edu (M.A.R.B.); michael.kjelland@mayvillestate.edu (M.K.); 2Department of Mechanical Engineering, North Dakota State University, Fargo, ND 58102, USA; chad.ulven@ndsu.edu

**Keywords:** bioplastic, wheat bran, arabinoxylan, tensile strength, biodegradability

## Abstract

The milling process of wheat annually generates over 150 million tons of wheat bran (WB), which has significant potential for bioplastic production. However, the production of bioplastics from these resources has never been explored until now. Wheat bran (WB) polymer was evaluated for its potential as an environmentally friendly biodegradable plastic, exhibiting a tensile strength of 2.3 MPa, elongation exceeding 100%, and resistance to diluted mineral acids, thereby highlighting its suitability for food packaging and related applications. The biodegradable plastic films were prepared through a molding process that involved three steps: (1) extraction of arabinoxylan from wheat bran, (2) hydrolysis and plasticization with glycerol, and (3) blending with polyvinyl alcohol (PVA) in varying proportions. The resulting bioplastic exhibits competitive mechanical properties and biodegradability. Furthermore, the biodegradable plastic developed in this research contributes to agricultural waste management, the development of value-added products, and the reduction of carbon emissions incurred from plastic industries. Additionally, it can replace and reduce reliance on synthetic plastics, which are non-degradable and a source of severe environmental pollution.

## 1. Introduction

The utilization of biomass recovered from inedible agricultural wastes for bioplastic production is a promising research avenue, offering significant environmental benefits within a circular economic framework [1,2,3,4]. Bioplastics are defined as biodegradable or nonbiodegradable polymers produced from renewable or natural biological sources. However, non-biodegradable bioplastics from biomass and fossil sources fundamentally follow a linear economy model and are not considered to be sustainable for example: bio-PE (bio-based polyethylene), bio-PET (bio-based polyethylene terephthalate) [5,6,7]. In contrast, biodegradable bioplastics are derived from biomass and support a circular economy by combining short-term biodegradability with the conversion of waste biomass into value-added products for example: starch-based plastic, cellulose-based plastic [8,9,10,11].

Bioplastics manufactured from plant biomass provides new functionalities to the resulting product, including antioxidant, antimicrobial, and nutraceutical properties [12]. During bioplastic production, biomass is typically dried and ground into powder form to reduce bacterial and enzymatic activities, helping to preserve its composition during storage [13]. Due to its lack of thermoplasticity, this dried and ground biomass cannot be directly converted into bioplastic. Therefore, several methods have been recommended for the digestion and dissolution of the biomass [2,4,14]. Among these, acidic and alkaline hydrolysis are the most used processes [2]. The resulting plant residue is plasticized with various plasticizers, including glycerol, glycol, xylitol, sorbitol, sugars, amides, urea, formamide, and ethylene-bis-formamide [15], and is blended with synthetic polymers like polyvinyl alcohol and polylactic acid to enhance both physicochemical properties and biodegradability [16].

Wheat is one of the most consumed crops in the world [17,18]. Wheat belongs to the genus *Triticum spp*. The wheat kernel contains 14.5% bran, which is removed during the milling process. Each year, a large quantity of wheat bran is produced, yet the majority is directed toward low-value uses such as cattle feed, with only about 10% incorporated into human food products. Due to limited market demand and logistical challenges, a significant portion remains underutilized or is discarded, often lead to environmental pollution [19,20,21].

Wheat bran contains carbohydrates like cellulose, hemicellulose, and lignin, as well as proteins, minerals such as calcium and phosphorus, silica, acid detergent fibers, and ash. Numerous studies have explored the production of value-added products such as bioplastics, biochar, and others from wheat bran fiber and/or wheat bran starch [22,23,24,25]. In this study, arabinoxylan, a major polysaccharide, was extracted from wheat bran and used for bioplastic.

Physico-mechanical properties are essential for evaluating the suitability of bioplastics as sustainable alternatives to conventional plastics. Among these properties, tensile strength, elongation at break, Young’s modulus, water solubility, and thermal and chemical resistance significantly influence the functionality and application of bioplastics in fields like food packaging, agriculture, and biomedical devices [26]. For example, tensile strength and elongation provide insights into the material’s ability to withstand mechanical stress and deformation, which are crucial for load-bearing or flexible packaging applications [27].

To assess these characteristics, a range of analytical techniques is employed. The Fourier-Transform Infrared Spectroscopy (FTIR) is used to detect chemical functional groups, while Scanning Electron Microscopy (SEM) reveals surface morphology and structural integrity, and tensile testing is evaluated by a Universal Testing Machine (UTM). Additionally, water contact angle (WCA) analysis helps determine hydrophilicity or hydrophobicity. Together, these techniques provide a comprehensive understanding of bioplastic performance and serve as essential tools in material design and improvement.

However, this study focuses on utilizing arabinoxylan derived from wheat bran to develop eco-friendly bioplastics, aiming to reduce the use of commercially available synthetic plastic.

## 2. Materials and Methods

### 2.1. Chemicals and Reagents

Glycerol, polyvinyl alcohol (PVA), and CuSO_4_.5H_2_O, commercially known as blue vitriol, used in this experiment were purchased from Thermo Scientific, Waltham, MA, USA. Potassium hydroxide (KOH) was obtained from Fisher Chemicals, Gell, Belgium. Sulfuric acid (H_2_SO_4_) was bought from Fisher Chemical, Fairlawn, NJ, USA.

### 2.2. Procedures

#### 2.2.1. Wheat Bran Collection and Extraction of Arabinoxylan

Wheat bran was collected from North Dakota Mill, Grand Forks, North Dakota. Upon collecting, the wheat bran was sieved to obtain a uniform size distribution of 2–3 mm of the bran by using a Hamilton Beach blender (Model 609-4, Hamilton Beach Inc., Glen Allen, VA, USA). Arabinoxylan was extracted following the method described in our previous work. A total of 40 g of wheat bran was treated with 400 mL of 4.5% potassium hydroxide (KOH) solution and stirred at 100 °C for 2 h. The resulting slurry was centrifuged at 8885× *g* for 20 min, and the supernatant was collected. To this, twice the volume of 95% ethanol was added, and the mixture was stored at 4 °C for 16 h to allow complete precipitation of arabinoxylan (AX). The precipitate was then recovered by centrifugation under the same conditions and thoroughly washed with distilled water to eliminate residual ethanol and excess KOH. Finally, the AX was dried in a vacuum oven at 40 °C for 24 h, followed by lyophilization at −50 °C using a freeze-dryer (Modulyod, Thermo Electron Corporation, Waltham, MA, USA). The schematic presentation of the extraction process is presented in Figure 1 [28,29].

#### 2.2.2. Bioplastic Synthesis

The bioplastic synthesis was conducted using a three-step process, following a slightly modified method described by Mendes et al. [30]. The first step involved extracting arabinoxylan. The second step includes hydrolysis and plasticization with glycerol. In this phase, arabinoxylan was placed in a beaker containing distilled water at a ratio of 1:20 (*w*/*v*). The mixture was heated to 80 °C for 20 min, after which glycerol was added as a plasticizer to enhance the flexibility and moldability of the plastic. Finally, the mixture was blended with polyvinyl alcohol (PVA) and stirred continuously at 80 °C for an additional two hours until the solution was thickened and turned into a jelly-like slurry. The thick slurry solution was poured into a Petri dish and dried overnight in an oven (Thermo Scientific, Heratherm OGS60, made in Langenselbold, Germany) at 60 °C. The following day, the Petri dish was removed from the oven while a bioplastic film was formed. The film was then allowed to cool to room temperature and pressed into sheets using a hydraulic hot press mold at 2000 Hg (mm) and 218 °F. The schematic diagram of the process is shown in Figure 2. During the process, a specific ratio of materials—arabinoxylan, glycerol, and polyvinyl alcohol (PVA)—was used, as detailed in Table 1.

The development of the biodegradable films based on polysaccharide–synthetic polymer blends has attracted growing attention because of their potential as sustainable alternatives to traditional plastics. Among these systems, arabinoxylan (AX), polyvinyl alcohol (PVA), and glycerol are often combined to create bioplastic films with enhanced mechanical, thermal, and barrier properties. The key interactions that support the compatibility and performance of these films are mainly driven by the extensive hydrogen bonding network formed among the three components. Arabinoxylan is a hemicellulose made up of a β-(1→4)-linked Xylan backbone with α-L-arabinofuranose side chains, both rich in hydroxyl groups that can act as hydrogen bond donors and acceptors. Likewise, PVA is a synthetic polymer consisting of repeating –CH_2_–CHOH– units, which provide a higher density of hydroxyl groups along its chain. When AX and PVA are blended, their hydroxyl groups form intermolecular hydrogen bonds, creating a physically cross-linked network that improves miscibility and reduces phase separation. These interactions are especially important because polysaccharides like AX are naturally hydrophilic but tend to produce brittle films on their own; adding PVA strengthens the hydrogen-bonded network, enhancing mechanical strength and film durability. The addition of glycerol, a small polyol with three hydroxyl groups, further influences the intermolecular bonding within the AX–PVA system. Glycerol acts as a plasticizer by breaking the strong inter- and intra-chain hydrogen bonds typically found in PVA and AX, which increases the mobility and flexibility of the polymer chains. Through hydrogen bonding, glycerol functions as a molecular bridge, interacting with hydroxyl groups on AX and PVA chains simultaneously (Figure 3). This bridging effect not only reduces the brittleness of the films but also makes them more processable. However, the arrangement of hydrogen bonds within the three blends can be seen as a dynamic balance, where AX–PVA bonds form the structural backbone, and glycerol-mediated bonds control flexibility and ductility. Figure 3 shows the hydrogen bonding interactions among them. Besides hydrogen bonding, weaker van der Waals forces and entanglements between AX and PVA chains also contribute to film stability. The combined interactions among AX, PVA, and glycerol lead to improved mechanical and barrier properties of the films [31,32,33,34].

Overall, the hydrogen bonding network among AX, PVA, and glycerol is crucial in determining the final properties of the bioplastic films, emphasizing the significance of molecular interactions in designing renewable packaging materials.

### 2.3. Characterization of the Synthesized Bioplastics

#### 2.3.1. Fourier Transform Infrared (FTIR) Spectroscopy

FTIR spectral analysis was conducted after extracting arabinoxylan from wheat bran. The FTIR spectra of the bioplastic film was analyzed after casting, and after two months of burial of the bioplastic film in soil that used for testing biodegradability. The FTIR analysis experiment was conducted at the Department of Chemistry, University of North Dakota, USA, using an FTIR machine model: NICOLETiS5, Thermo Scientific, Waltham, MA, USA. The spectra, recorded in the 4000–650 cm^−1^ range, displayed absorption frequency against transmittance (%). Data acquisition resulted in an incidence angle of 45°, a 2 mm sampling area, 16 background scans, an optical resolution of 0.8 cm^−1^, and data spacing of 0.06 cm^−1^. The degree of cationization was quantified using the following equation:(1)C=(I1648−I1495)/I1648×100%,
where I1648 and I1495 represent the peak intensities at 1648 cm^−1^ and 1495 cm^−1^, respectively.

#### 2.3.2. Mechanical Properties

Tensile strength (TS) was determined using an Instron Model 5542, made in the Norwood, MA, USA, following ASTM D412 [35]. Bioplastic film specimens were cut to 2.7 mm in width, with an average gauge length of 50 mm, and were mounted on the testing apparatus. The tensile test was performed at a crosshead speed of 20 mm/min to evaluate the elastic modulus. Elongation at break (EAB) was calculated by assessing the ratio of the specimen’s final length after failure to its original length under applied tensile stress.

#### 2.3.3. Scanning Electron Microscope (SEM) Analysis

The surface morphology of the bioplastic was observed using high-resolution Scanning Electron Microscopy at the Department of RCA Electron Microscopy Core, North Dakota State University, USA. First, the treated plastic samples were cut with scissors and affixed to cylindrical aluminum stubs using silver paint (SPI Products, West Chester, PA, USA). A thin conductive layer of gold was then applied to the mounted samples with a sputter coater (Cressington 108Auto, Ted Pella, Redding, CA, USA). Imaging was conducted with a JEOL JSM-6490LV scanning electron microscope (JEOL USA, Inc., Peabody, MA, USA) at an accelerating voltage of 15 kV.

#### 2.3.4. Film’s Thickness Measurements

The thickness of the film samples was measured by using a digital micrometer with a sensitivity of 0.01 mm, by taking measurements at five randomly selected points. The mean thickness value was then used for subsequent mechanical and optical property analyses.

#### 2.3.5. Film’s Transparency

The transparency of the film samples was evaluated following the slightly modified method described by Mulyono et al. [36]. Bioplastic films were cut into dimensions of 1 cm × 3 cm to fit the width and height of a standard cuvette. Each film sample was attached to the side of the cuvette, and a synthetic polyethylene film served as the control. Absorbance measurements were taken at a wavelength of 800 nm by using a GENESYS 10S UV-VIS Spectrophotometer, Thermo Scientific, USA, at Mayville State University. Blue vitriol solution was used for the measurement of absorption. The transmittance (%T) was then calculated using the following equation:(2)% of T=antilog (2−absorbance).

And transparency was calculated by using the following formula:(3)$$Transparency=\frac{log\;\%\;T}{b},$$
where *%T* is the transmittance at 800 nm and b is the thickness of the bioplastic film in millimeters.

#### 2.3.6. Film’s General Appearance

The visual characteristics of the bioplastic films were evaluated through systematic observation to assess surface texture, uniformity, color, opacity, and the presence of any physical defects such as bubbles, cracks, or warping. Each sample was photographed under standardized conditions—consistent ambient lighting, neutral background, and fixed camera settings—to ensure reliable visual documentation. These images were used to compare the esthetic and structural integrity of films prepared under different processing conditions or compositions. Visual assessment served as an initial qualitative indicator of film quality and was used alongside analytical techniques to interpret the material’s overall performance and stability.

#### 2.3.7. Water Contact Angle

The way water interacts with the surface is commonly described by two phenomena: hydrophilicity and hydrophobicity. The water contact angle (WCA) is used as a key indicator to evaluate this behavior. Over the years, extensive research has utilized this method to investigate and understand the surface properties [37,38,39,40,41]. In this study, the water contact angle of the synthesized film was measured to evaluate the wearability of the film. A pluggable USB 2.0 Digital Microscope (purchased from amazon.com) was employed to capture images of water droplets placed on the surface of the film samples. A Ziploc bag and a polyethylene bag collected from local Walmart were used as a reference to compare the water contact angle of the synthesized film. The experiment was conducted at an ambient temperature of 23 °C, a needle width of 0.525 mm, and a droplet volume of 5 µL. The water contact angle was measured using ImageJ 1.54g software [42]. Images were taken multiple times for each film, and the best image was selected for analysis.

#### 2.3.8. Water Absorption Percentage

All prepared bioplastic films were cut into dimensions of 2 cm × 2 cm and oven-dried at 60 °C for 24 h. The initial mass of each film (*M_0_*) was recorded. The samples were then immersed in 50 mL of distilled water for 24 h at room temperature. After immersion, the films were removed, gently wiped with Pacific Blue Select Multifold Premium 2-Ply Paper Towels by GP PRO (Georgia-Pacific) to remove surface water and weighed to determine the final mass (*M_1_*). The water absorption percentage was calculated using the following formula:(4)$$Water\;Absorption=\frac{M1-Mo}{Mo}\;\times 100\%,$$
where *Mo* and *M1* are the initial (dry) mass and the water immersion mass of the film, respectively. The experiment was conducted by the modified method described by Saberi et al. [43].

#### 2.3.9. Effect of Acid

Accurately weighed samples of about 1.0 g of the synthesized bioplastics from arabinoxylan were immersed in sulfuric acid (H_2_SO_4_) solutions with concentrations of 10%, 20%, 30%, and 40%. The samples were continually dried and weighed over 2 days interval for 10 days to assess weight loss, and a photograph was taken after each data collection interval. The experiment was conducted by following the method described by Mostafa et al. [44].

#### 2.3.10. Effect of Alkalis

Accurately weighed samples (synthesized bioplastics from arabinoxylan) of about 1.0 g were taken by using a high-accuracy electronic balance (Denver Instruments XE-100, USA, Serial No NO111601, Max: 100 g, d: 0.0001 g). The plastic pieces were immersed in potassium hydroxide (KOH) solutions at varying concentrations of 10%, 20%, 30%, and 40%. The percentage of weight loss was calculated every second day for ten days, and a photograph was taken after each data collection interval.

#### 2.3.11. Biodegradability Test

The biodegradability experiment was conducted following modified procedures described by various researchers [44,45]. The synthesized bioplastic samples were oven-dried at 45 °C for two hours and weighed on a high-accuracy electronic balance. The weighted bioplastic pieces were buried in natural soil obtained from a fallow land surrounding Mayville State University to assess biodegradation. Throughout the experiment, soil moisture content and temperatures were carefully maintained. At four-week intervals, the samples were washed, oven dried at 60 °C, and weighed to calculate the percentage of weight loss, with a photograph taken after each data collection. For the final data collection, a Scanning Electron Microscope (SEM) image and a Fourier Transform Infrared (FTIR) spectrum were assessed with a photograph for general appearance.

## 3. Results

### 3.1. Fourier Transform Infrared (FTIR) Spectroscopy of WBAX and WBAX Bioplastic

Figure 4a,b shows the FTIR spectra of arabinoxylan extracted from wheat bran and the bioplastic made from it, respectively. From Figure 4a, the characteristic spectrum for arabinoxylan was observed at 3205 cm^−1^, 2918 cm^−1^, and 1018 cm^−1^, which correspond to O-H stretching, C-H stretching, and C-OH stretching of glycosidic linkages, respectively. Figure 4b indicates the characteristic peaks of WBAX 1:2:1 bioplastic.

The Fourier Transform Infrared (FTIR) spectrum of the WBAX 1:2:1 bioplastic film provides direct evidence of molecular interactions among arabinoxylan, polyvinyl alcohol (PVA), and glycerol, confirming the successful formation of a hydrogen-bonded polymeric network. Figure 4b shows the characteristic absorption peaks located at 3252 cm^−1^, 2911 cm^−1^, 1651 cm^−1^, 1419 cm^−1^, 1039 cm^−1^, and 676 cm^−1^. The broad peak centered at 3252 cm^−1^ corresponds to the stretching vibrations of hydroxyl (–OH) groups, which are abundant in arabinoxylan, PVA, and glycerol. The broadness and intensity of this band reflect the presence of extensive intermolecular hydrogen bonding, a key feature that stabilizes the three bioplastic matrices. The shift in this band compared to pure AX or PVA spectra reported in earlier studies suggests that new hydrogen-bonded interactions are formed upon blending, thereby confirming miscibility and compatibility of the components.

The absorption band observed at 2911 cm^−1^ is assigned to aliphatic C–H stretching vibrations, primarily arising from the methylene (–CH_2_–) groups of PVA and the arabinose side chains of AX. The intensity of this band provides evidence of the integration of the synthetic polymer backbone within the polysaccharide-rich film, indicating the formation of a composite material rather than a physical mixture. The peak at 1651 cm^−1^ is particularly significant as it is attributed to the C=O stretching of triglyceride linkages. This band confirms the chemical contribution of glycerol within the polymer network and highlights its role as a plasticizer. The incorporation of glycerol facilitates flexibility by reducing intermolecular cohesion among AX and PVA chains, but its presence in the FTIR spectrum simultaneously verifies its molecular-level participation in the bioplastic structure.

The peak at 1419 cm^−1^ is commonly associated with the bending vibrations of –CH_2_ groups and may also reflect skeletal vibrations within the polysaccharide backbone. Its presence suggests that both carbohydrate and synthetic polymer chains are structurally integrated within the film. The sharp band at 1039 cm^−1^ is attributed to C–O stretching vibrations and glycosidic linkages within arabinoxylan, serving as direct evidence that the polysaccharide backbone remains intact during film formation. Importantly, the spectral region below 1039 cm^−1^ displays characteristic absorption associated with strong hydrogen bonding of hydroxyl groups. This region is critical because it confirms that the hydroxyl functionalities of AX, PVA, and glycerol are not acting independently but are instead engaged in extensive intermolecular interactions, producing a dense and cohesive hydrogen-bond network.

Therefore, these FTIR features provide conclusive evidence that the WBAX 1:2:1 blend does not represent a simple physical mixture of its components but rather forms a unified polymeric matrix. The broad hydroxyl stretching band, the presence of triglyceride-related peaks, and the absorption features below 1039 cm^−1^ collectively confirm the molecular interactions necessary for the formation of a flexible, stable, and homogeneous bioplastic film. Such spectral signatures are consistent with previous reports on polysaccharide–PVA–glycerol films and strongly support the role of hydrogen bonding as the primary driving force for structural integrity and film formation [46].

### 3.2. Mechanical Properties of Synthesized Bioplastic

Tensile stress (TS), tensile strength, tear resistance and elongation at break (EAB) are essential indicators of a film’s mechanical performance, reflecting its ability to maintain structural integrity under applied stress. The wheat bran extracted arabinoxylan-based bioplastic films of different ratios (WBAX) demonstrated stress at break ranging from 1.2 MPa to 3.34 MPa, tensile strength ranging from 0.83 to 2.3 MPa and an elongation at break (EAB) across from 66% to 281% (Figure 5). The relatively higher stress and strength values were attributed to the strong interactions between arabinoxylan, polyvinyl alcohol, and glycerol, as a plasticizer and reinforcing agent, thereby enhancing its rigidity and structural stability. However, among the four different bioplastic compositions, the WBAX bioplastic obtained from a matrix blend ratio of 1:2:1 showed the best mechanical performance, with a stress at break of 3.34 MPa, tensile strength of 2.3 MPa and an EAB of 137%. This performance is comparable to that of low-density polyethylene (LDPE) and cotton fiber-based bioplastics [47].

In addition to tensile stress, tensile strength, and elongation at break, tear resistance plays a crucial role in evaluating the mechanical reliability of bioplastic films. The WBAX films demonstrated a trend where higher tensile stress and strength values correlated with enhanced resistance to tear propagation, as stronger intermolecular interactions between arabinoxylan, PVA, and glycerol reduced crack initiation and growth. For example, the 1:2:1 WBAX composition, which exhibited the highest stress at break (3.34 MPa), tensile strength (2.3 MPa), and a balanced EAB (137%), also presented superior tear resistance compared to other formulations. This suggests that the optimized matrix not only withstands uniaxial stretching but also resists progressive tearing forces, a property critical for practical applications such as flexible packaging, where films are subjected to puncture and shear stresses.

### 3.3. Scanning Electron Microscope (SEM) Analysis of Bioplastic

Figure 6 shows the surface morphology of WBAX 1:2:1 bioplastic at two different resolutions. The SEM image displays the smooth surface of the bioplastic, although it contains some micro-voids. These micro-voids are due to the degree of glycerol dispersion within the plastic matrix. During the gelatinization process, the hydrogen bonds in the long arabinoxylan chains break down, allowing water molecules to penetrate the hydroxyl groups of the arabinoxylan, resulting in the formation of voids and micro-voids. The surface morphology of this bioplastic is found to be similar to the morphology of the polyethylene elastic material studied by Gere et al. [48].

### 3.4. Film’s Transparency

Table 2 shows that the bioplastic films exhibited lower transparency compared to synthetic polyethylene, which may be due to the presence of fillers and the integral thickness of the bioplastic films. Nonetheless, the PAV-based bioplastic demonstrated superior transparency relative to results reported in previous studies. For instance, Mulyono et al. [24] reported a maximum transparency value of 3.13 for tapioca-based films, whereas the bioplastic films developed in this study achieved a transparency value of 1.50.

### 3.5. Water Contact Angle

Figure 7 illustrates the water contact angle measurements, accompanied by images of water droplets on the surfaces of the film samples. Table 3 presents the water contact angle (WCA) values of the bioplastic film developed in this study, alongside those of a commercially available synthetic plastic bag and selected literature references [49]. The results indicate that the WBAX bioplastic film exhibits a water contact angle comparable to that of the market plastic, suggesting similar surface wettability characteristics. However, the WCA value of the WBAX bioplastic film remains below 90°, indicating that it is not truly hydrophobic. For a material to be classified as hydrophobic, its water contact angle must exceed 90°. Therefore, while the WBAX film demonstrates reduced water affinity relative to other bioplastics, it does not meet the threshold for hydrophobicity, implying potential applications where moderate water resistance is sufficient.

### 3.6. Effect of Acids on Bioplastic

The concentration of sulfuric acid has an impact on the weight loss behavior of the bioplastic produced from wheat bran-extracted arabinoxylan. As shown in Figure 8a, under acidic conditions, the WBAX bioplastic showed increasing weight loss with higher concentrations of sulfuric acid, with complete or near-complete dissolution observed at 30–40% acid within 2 days and 20% acid within 4 days. The enhanced degradation at higher acid concentrations can be explained by acid-catalyzed hydrolysis of the glycosidic bonds in arabinoxylan. The arabinoxylan polymer consists of a β-(1→4)-linked xylose backbone with arabinose side chains; in strongly acidic media, the protonation of oxygen atoms in the glycosidic bonds increases their susceptibility to cleavage, leading to chain scission and solubilization of the polymer matrix.

The partial residues observed even after extensive acid exposure indicate that not all polymer chains are equally accessible or reactive, possibly due to crystalline regions, cross-linked areas, or interactions with glycerol plasticizer that confer localized resistance. Such residual material forms a thicker, slurry-like solution rather than complete dissolution. The observed acid resistance at low concentrations reflects the inherent stability of arabinoxylan in mildly acidic conditions, which is advantageous for applications where exposure to mild acids may occur. Compared to commercial cellulose acetate (CA), WBAX bioplastics demonstrate slightly superior acid resistance, likely due to the presence of cross-linking and hydrogen bonding among arabinoxylan chains and glycerol, which stabilize the polymer network [50].

### 3.7. Effect of Alkalis on Bioplastic

Figure 9 shows the weight loss behavior of bioplastic, produced from wheat bran-extracted arabinoxylan, when treated with different concentrations of potassium hydroxide (KOH) for ten days.

When exposed to alkaline environments, WBAX bioplastic demonstrated strong resistance, with lower weight loss at higher KOH concentrations (maximum 52% at 10% KOH, decreasing at 40% KOH). Alkali-induced degradation primarily involves base-catalyzed hydrolysis of ester or ether linkages and cleavage of acetyl or other substituent groups. Unlike acidic hydrolysis, which targets glycosidic bonds, alkaline hydrolysis is slower and may be limited by steric hindrance, cross-linking, or intra/intermolecular hydrogen bonding within the biopolymer network. This can explain why WBAX films retain a significant portion of their mass even under strong alkali exposure.

Interestingly, the higher resistance to alkali compared to CA is likely due to the flexible arabinoxylan network, which allows stress distribution and limits polymer chain breakdown. Additionally, glycerol and other plasticizers can act as protective agents by stabilizing hydrogen bonding and reducing water penetration, which slows alkali-mediated hydrolysis [51].

The differing behaviors in acidic versus alkaline environments are fundamentally due to the mechanisms of hydrolysis and the chemical structure of arabinoxylan:

Acidic hydrolysis primarily cleaves glycosidic bonds, resulting in rapid depolymerization and solubilization at higher acid concentrations. Protonation accelerates bond cleavage, making the polymer more vulnerable to acids. Alkaline hydrolysis targets ester or ether linkages and substituents rather than the main backbone. Strong hydrogen bonding, cross-linking, and steric effects reduce polymer accessibility, leading to slower and less extensive degradation compared to acids [52].

Thus, WBAX bioplastics exhibit high acid sensitivity at elevated concentrations but strong alkali resistance, indicating that the polymer network is more susceptible to proton-mediated cleavage than base-mediated degradation. These characteristics make WBAX bioplastics particularly suitable for applications where moderate acid exposure may occur, but alkaline resistance is required, such as in certain food packaging or agricultural films.

### 3.8. Water Absorption Percentage of Bioplastics

Film water absorption tendency indicates the presence of hydrophilic components in the material. Figure 10a illustrates the water absorption characteristics of the bioplastic film. The bioplastic film’s increased water absorption is due to the presence of PVA, which has water-absorbing qualities. In contrast, those films exhibited a lower tendency, indicating the strong intermolecular forces and cross-linking within the matrix, limiting their interaction with water. Consequently, the bioplastic film at a 1:2:4 ratio displayed the lowest water solubility, while the film at a 1:2:1 proportion showed a moderate level of solubility.

Figure 10b illustrates the water solubility behavior of the bioplastic films. Among the tested formulations, the WBAX film with a 1:2:1 ratio exhibited the lowest water absorption, with solubility below 40%. In contrast, the film with a 1:2:4 ratio demonstrated the highest solubility, reaching approximately 70%. The reduced solubility observed in the 1:2:1 film suggests stronger intermolecular interactions and a more compact polymer matrix, which limits water penetration and absorption.

The water solubility of bioplastic films is intrinsically linked to the chemical composition, molecular interactions, and microstructure of the polymer matrix. In this study, the bioplastic films were prepared with varying ratios of wheat bran-extracted arabinoxylan (WBAX), polyvinyl alcohol (PVA), and glycerol, which directly influenced their solubility behavior.

Among the formulations, the WBAX:PVA: Glycerol films with a 1:2:1 ratio exhibited the lowest solubility (<40%), whereas the 1:2:4 films showed the highest solubility (~70%), with intermediate values observed for the 1:2:2 formulation. These differences can be attributed to several molecular and structural factors [53,54,55]:

Hydrophilic Component Content: PVA is highly hydrophilic due to the presence of hydroxyl (−OH) groups capable of forming hydrogen bonds with water. Films containing higher amounts of PVA and glycerol display increased water affinity, facilitating water penetration and dissolution. In contrast, films with lower hydrophilic content limit the availability of water-interactive sites, reducing solubility.

Intermolecular Interactions: Strong hydrogen bonding between WBAX, PVA, and glycerol can form a densely cross-linked network within the polymer matrix. This compact structure restricts water molecules from diffusing into the film, thereby decreasing solubility. The 1:2:1 formulation likely possesses optimal cross-link density, producing a rigid matrix that limits water uptake, while the 1:2:4 film, with excess glycerol, disrupts these interactions, resulting in a looser, more hydrophilic matrix.

Plasticizer Effect of Glycerol: Glycerol acts as a plasticizer by increasing the mobility of polymer chains and reducing intermolecular forces. While low glycerol content preserves matrix integrity, higher glycerol levels enhance chain flexibility and free volume, allowing water molecules to penetrate more easily, thereby increasing solubility. The 1:2:4 formulation exhibits this effect, with glycerol-induced plasticization dominating the network stability.

Matrix Compactness and Morphology: The degree of polymer packing within the film influences water accessibility. A densely packed film with extensive intermolecular interactions presents fewer voids and channels for water diffusion, while a less compact film with disrupted hydrogen bonding and increased free volume permits higher water uptake. SEM studies in similar WBAX/PVA films have shown that higher glycerol content results in a more porous morphology, consistent with the observed increase in solubility.

Synergistic Effects of Composition: The balance between WBAX, PVA, and glycerol determines the overall hydrophilic–hydrophobic nature of the film. Formulations with excessive hydrophilic components (high PVA/glycerol ratios) favor water absorption and solubility, whereas those with optimized WBAX-to-PVA ratios promote stronger intermolecular cohesion, limiting water interaction.

In summary, the variation in water solubility among the bioplastic films is primarily governed by the interplay of hydrophilic content, hydrogen bonding, plasticizer concentration, and matrix compactness. Films with higher PVA and glycerol content exhibit enhanced solubility due to increased hydrophilicity and chain mobility, while films with lower plasticizer content and higher cross-linking density resist water penetration, demonstrating reduced solubility.

### 3.9. Biodegradability Analysis of the Bioplastic

The biodegradability of the WBAX bioplastic film was evaluated through a soil burial testing for two months, and the structural changes were analyzed using Scanning Electron Microscopy (SEM) and Fourier Transform Infrared Spectroscopy (FTIR). In Figure 11a,b, the SEM micrographs revealed notable morphological changes on the film surface after degradation. Compared to the smooth and uniform structure of the undegraded film, the buried samples exhibited cracks, voids, and surface erosion, indicating microbial and environmental activity that contributed to the breakdown of the polymer matrix.

These surface disruptions are characteristic of ongoing biodegradation, where microbial enzymes and soil moisture gradually weaken the film structure. The presence of micro-voids supports the hypothesis that biodegradation initiates through localized enzymatic attack, which leads to fragmentation of the polymer chains and structural disintegration. The physical appearance of the bioplastic film before and after degradation has been displayed in Figure 11c,d.

Further confirmation of chemical degradation was obtained through FTIR analysis. Figure 12 shows a comparison of spectra before and after the two-month burial period, revealing significant changes in peak intensities, reflecting the breakdown of functional groups. Most notably, the absorption peak below 1000 cm^−1^, which corresponds to the hydrogen bonding among the matrix blend, disappeared entirely after degradation. The disappearance of these peaks indicates molecular degradation (hydrogen bond disruption) during the biodegradation process.

## 4. Discussion

The wheat bran-extracted arabinoxylan (WBAX)-based bioplastic film developed in this study exhibits a promising combination of mechanical performance, flexibility, chemical resistance, and environmental sustainability, highlighting its potential as a viable alternative to conventional synthetic plastics, particularly low-density polyethylene (LDPE), which dominates global packaging applications and contributes significantly to environmental pollution [56]. The incorporation of arabinoxylan derived from wheat bran, a widely available agro-industrial byproduct, underscores the potential for value-added utilization of agricultural residues in the production of eco-friendly materials.

Mechanically, the WBAX bioplastic demonstrated a tensile strength of 3.34 MPa as shown in Table 4, which, while lower than the typical range of commercial LDPE (10–30 MPa), aligns with the functional requirements of flexible packaging applications where high tensile strength is less critical [57]. Its elongation at break of 138% falls within the lower end of LDPE elongation ranges (100–600%), indicating adequate flexibility for practical use. This combination of moderate tensile strength and appreciable elongation suggests that WBAX films are particularly suitable for applications requiring pliability rather than load-bearing capacity, such as food wraps, disposable packaging, or lightweight protective films. These mechanical characteristics are comparable to other polysaccharide-based bioplastics reported in the literature [58,59,60,61], which often trade higher strength for increased flexibility and environmental compatibility.

Surface wettability analysis revealed a water contact angle (WCA) of approximately 80°, indicating moderate hydrophilicity. While this is below the typical hydrophobicity threshold of 90° observed in LDPE [62], it is sufficient to provide a reduced water affinity, making the film suitable for short-term storage or controlled moisture environments. Partial hydrophilicity also promotes better adhesion in multi-layer composite materials, which could be advantageous in food packaging or agricultural film applications. Importantly, the water solubility of WBAX bioplastic (~40%) highlights its partial water resistance while maintaining biodegradability, a crucial feature that differentiates it from LDPE, which is highly resistant to degradation and persists in the environment over extended periods [63].

From a chemical resistance perspective, WBAX films exhibited notable resilience in low concentrations of acids and high concentrations of alkalis, reflecting their potential suitability in chemically variable environments, such as agricultural mulches, food packaging, or industrial coatings. Furthermore, the transparency of WBAX films enhances their utility in applications where visual inspection or product presentation is important, such as packaging for fresh produce or consumer goods. This optical property, coupled with moderate mechanical and chemical stability, positions WBAX films as a functional and sustainable alternative for a range of applications [64,65,66].

Despite their advantages, WBAX bioplastics still exhibit limitations, particularly in comparison to LDPE, in terms of hydrophobicity and tensile strength. Enhancing the hydrophobic character through chemical surface modification, cross-linking strategies, or blending with other biopolymers could further expand their functional scope and water resistance. Additionally, mechanical reinforcement, potentially through the incorporation of natural fibers, nanocellulose, or plasticizers, may improve tensile properties without compromising biodegradability. Future studies could also explore multilayer film formation, where WBAX is combined with other biopolymers to achieve tailored mechanical, barrier, and hydrophobic properties, aligning with the performance expectations of conventional plastics while maintaining environmental benefits.

In conclusion, WBAX bioplastic films provide a promising balance of flexibility, moderate mechanical strength, partial water resistance, and biodegradability, offering a sustainable alternative to conventional synthetic polymers. While further optimization is needed to enhance specific performance metrics, their environmentally compatible profile, transparency, and chemical resistance demonstrate strong potential for adoption in packaging, agricultural, and other short-term-use applications. By addressing current limitations through material modification or composite formulation, WBAX bioplastics could play a significant role in reducing reliance on non-degradable plastics and mitigating environmental pollution.

## 5. Conclusions

The eco-friendly arabinoxylan-based bioplastic derived from wheat bran was successfully synthesized via a multi-step procedure and comprehensively characterized using various analytical and environmental assessment techniques. Among the tested formulations, the 1:2:1 blend ratio exhibited the best overall performance, demonstrating superior mechanical strength, flexibility, optical clarity, and moderate hydrophobicity, comparable to conventional LDPE and other reported biodegradable plastics. The material showed notable chemical stability, with high resistance to alkali and dilute mineral acid solutions, while retaining excellent biodegradability. This combination of durability and environmental compatibility underscores its potential to mitigate the ecological impact associated with petroleum-based plastics. The satisfactory functional and environmental properties of WBAX bioplastics indicate promising applications in packaging, food storage, and various plastic tools, highlighting their role as a sustainable, biodegradable alternative in modern material design.

## Figures and Tables

**Figure 1 polymers-17-02488-f001:**
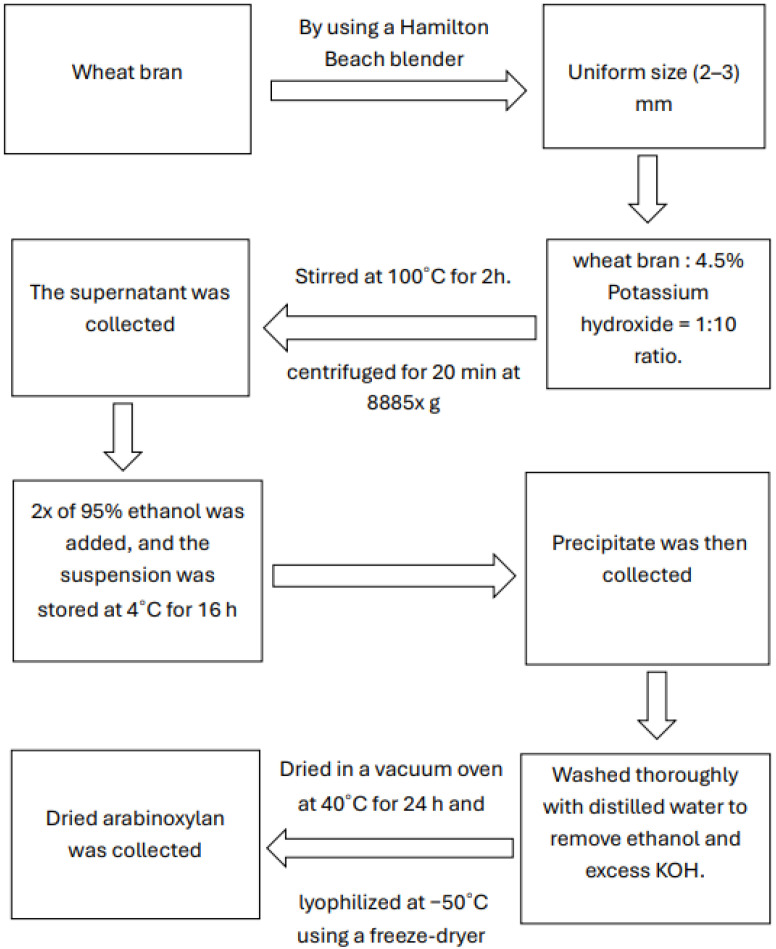
Flowchart of arabinoxylan extraction from wheat bran.

**Figure 2 polymers-17-02488-f002:**
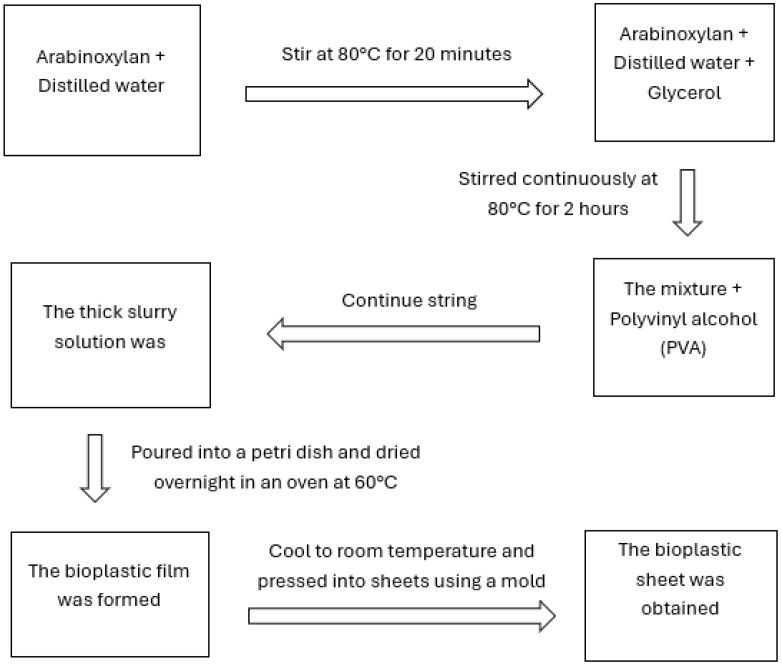
Schematic diagram of bioplastic synthesis.

**Figure 3 polymers-17-02488-f003:**
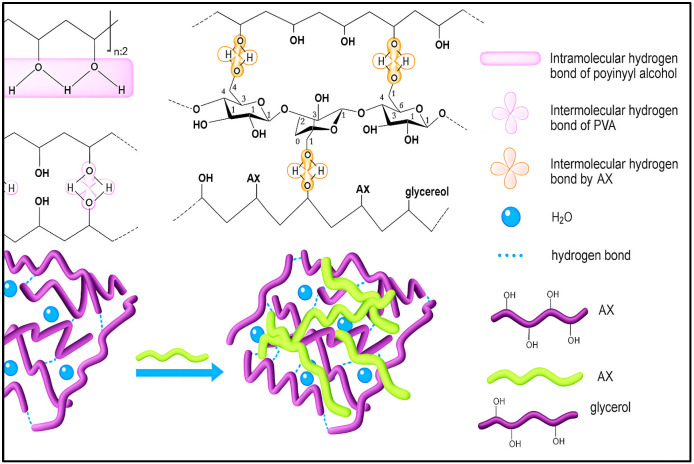
Hydrogen bonding interactions among PVA, AX, and glycerol.

**Figure 4 polymers-17-02488-f004:**
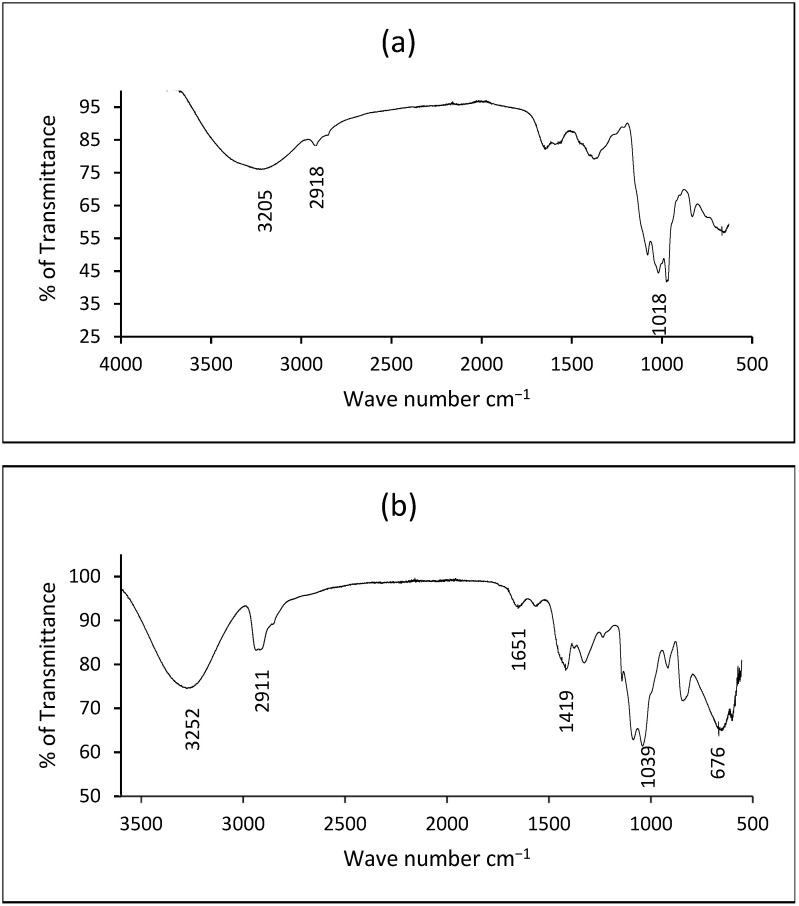
FT_IR spectra of (**a**) wheat bran extracted arabinoxylan, (**b**) WBAX 1:2:1 bioplastic.

**Figure 5 polymers-17-02488-f005:**
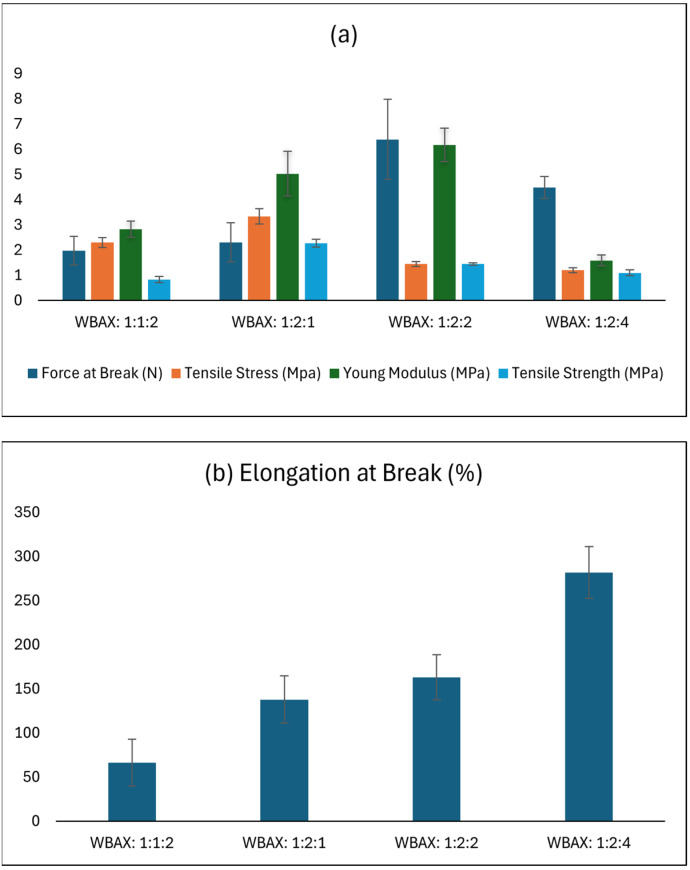
Mechanical properties of WBAX bioplastic at different ratios; (**a**) force at break, tensile stress, young modulus, tensile strength, (**b**) elongation at break.

**Figure 6 polymers-17-02488-f006:**
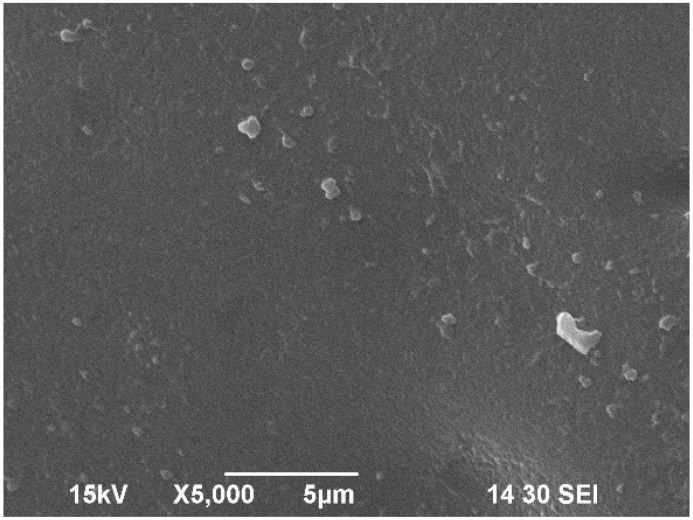
SEM image of WBAX 1:2:1 bioplastic at 5000× resolution.

**Figure 7 polymers-17-02488-f007:**
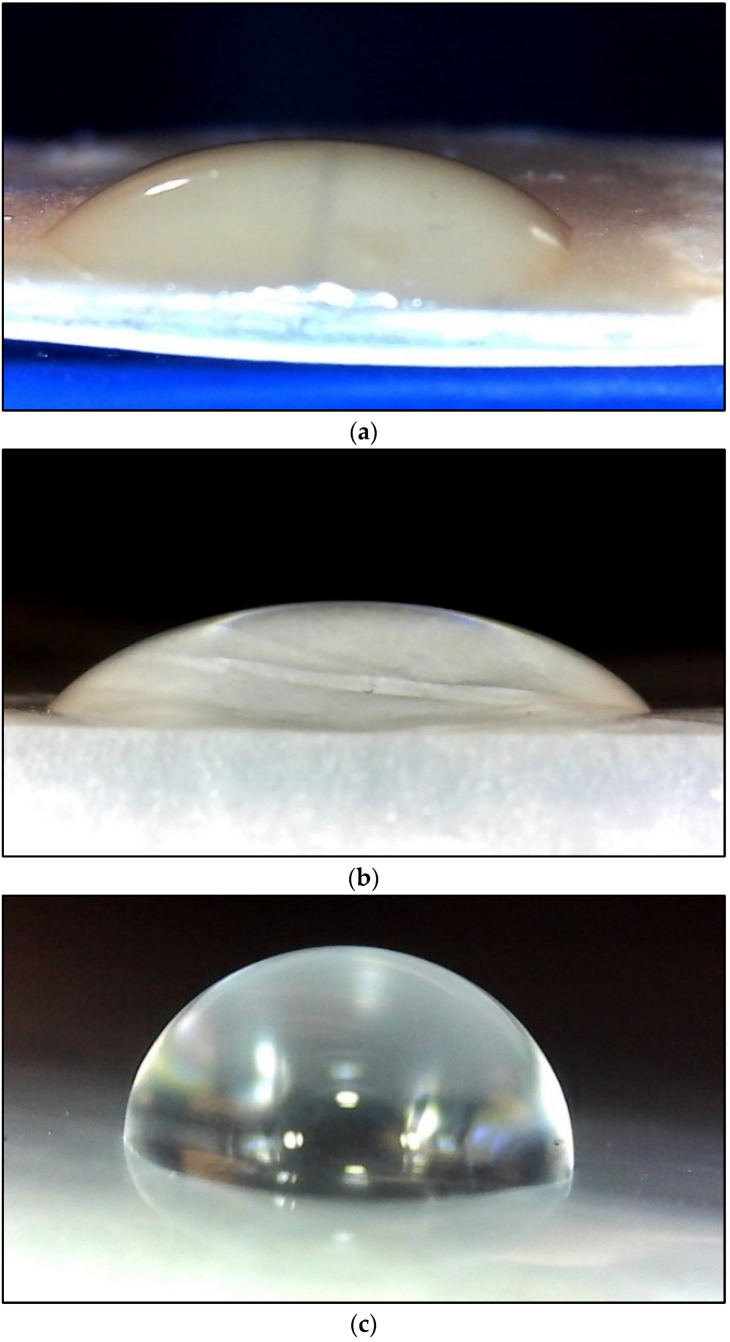
Water Contact Angle of (**a**) WBAX 1:2:1 bioplastic, (**b**) Walmart plastic bag, (**c**) Ziploc plastic bag.

**Figure 8 polymers-17-02488-f008:**
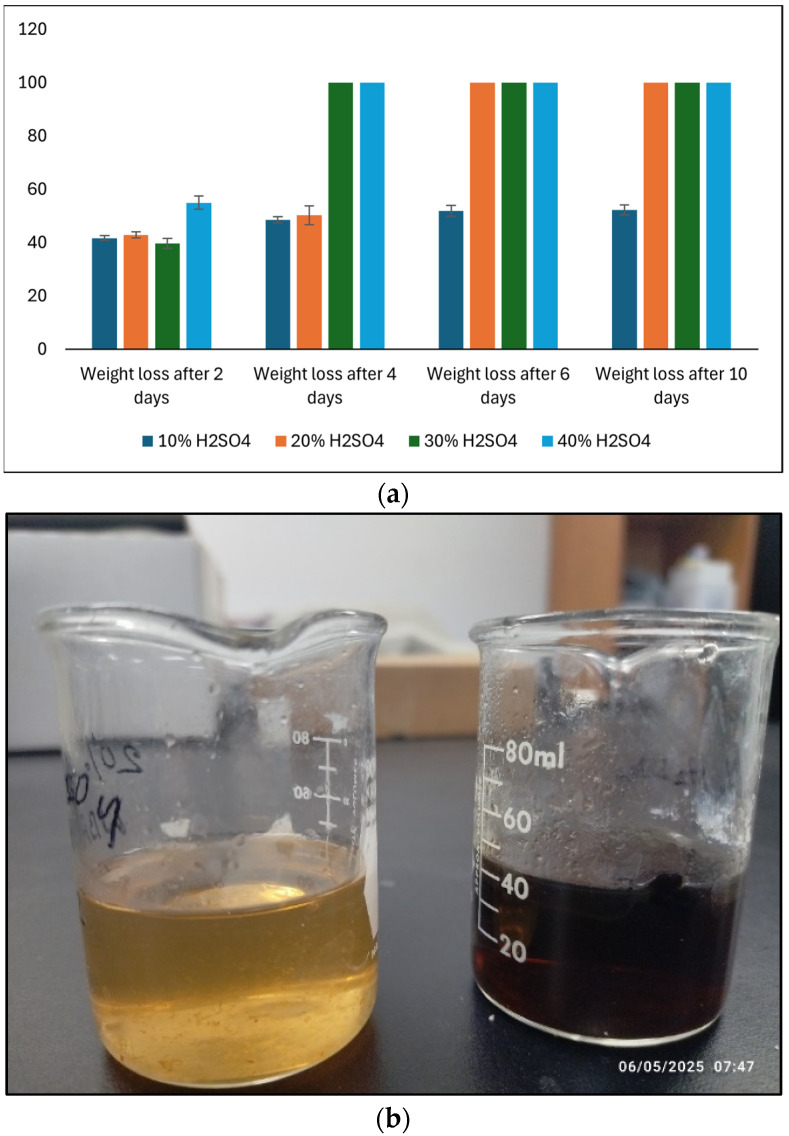
(**a**) Effect of acid concentration and treatment duration on weight loss, (**b**) the thicker and slurry solution after 2 days of dissolution (**right** side), and 40% acid solution (**left** side).

**Figure 9 polymers-17-02488-f009:**
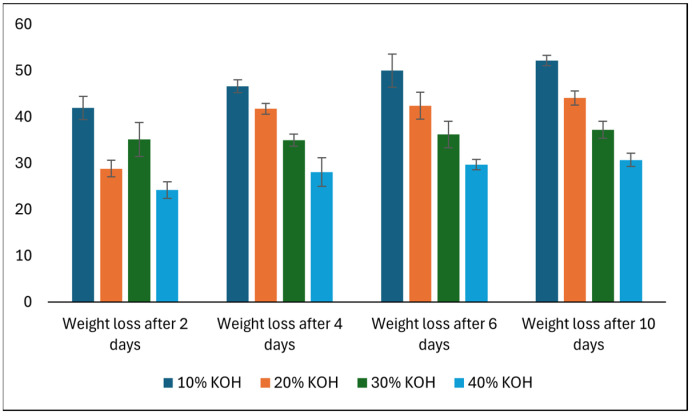
Effect of alkali concentration and treatment duration on weight loss.

**Figure 10 polymers-17-02488-f010:**
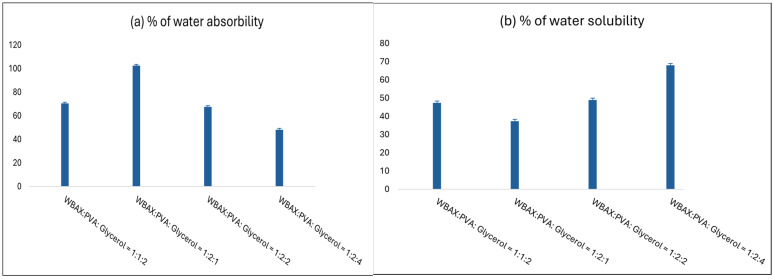
Water Absorption property of WBAX bioplastics (**a**) % of water absorbability, (**b**) % of water solubility.

**Figure 11 polymers-17-02488-f011:**
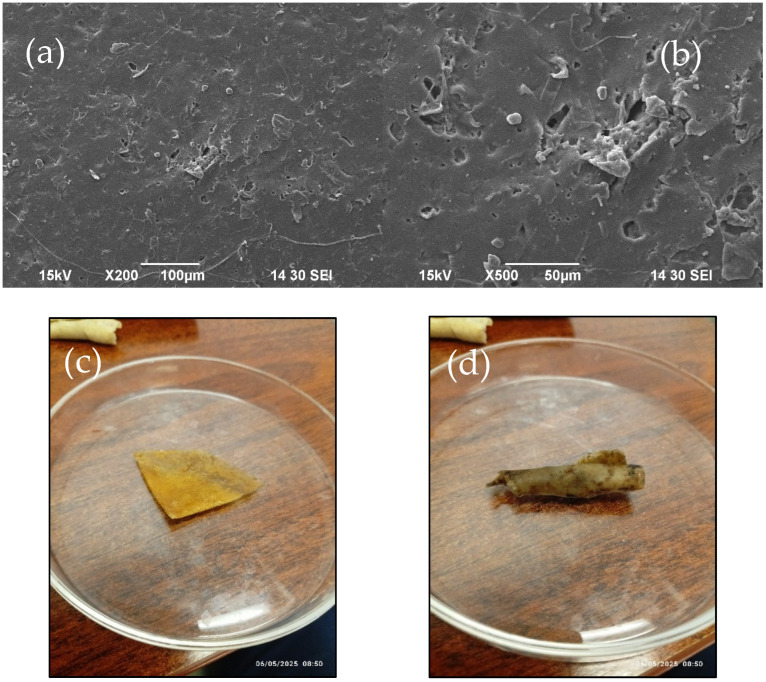
SEM image of (**a**) WBAX 1:2:1 after 2 months of natural degra-dation at 200× resolution, (**b**) WBAX 1:2:1 after 2 months of natural degradation at 500× resolution, (**c**) physical appearance of WBAX 1:2:1 before degradation, (**d**) physical appearance of WBAX 1:2:1 after degradation.

**Figure 12 polymers-17-02488-f012:**
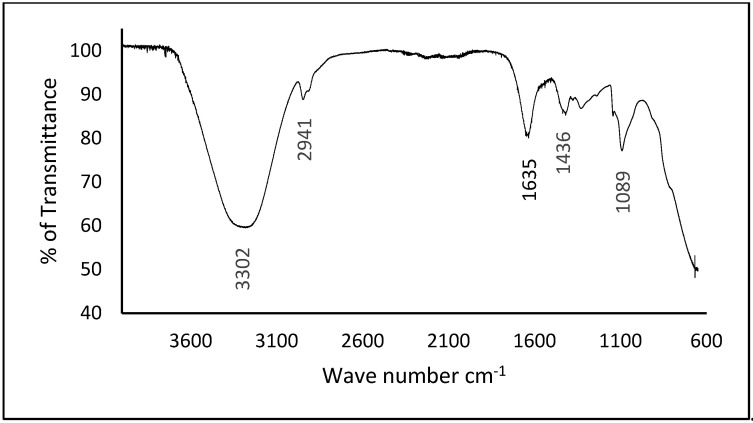
FTIR spectra of WBAX 1:2:1 bioplastic after 2 months of degradation.

**Table 1 polymers-17-02488-t001:** The composition ratio of the bioplastic blend.

Title	Wheat Bran Arabinoxylan (WBAX)	Polyvinyl Alcohol (PVA)	Glycerol
WBAX 1:1:2	1	1	2
WBAX 1:2:1	1	2	1
WBAX 1:2:2	1	2	2
WBAX 1:2:4	1	2	4

**Table 2 polymers-17-02488-t002:** Transparency of bioplastic films.

Film Type	Absorbance	Transmission%	Thickness	Transparency
WBAX bioplastic film	1.23	5.92	0.19	1.50
CPS bioplastic film	1.25	5.60	0.17	1.52
Ziploc plastic bag	0.05	89.23	0.02	3.57
Walmart plastic bag	-	-	-	3.78
tapioca-based films	-	-	-	3.13

**Table 3 polymers-17-02488-t003:** Water Contact Angle (WCA) of different plastics.

Film Type	Water Contact Angle (Degree)	SD
WBAX bioplastic film	75.80	0.60
Ziploc plastic bag	124.83	1.11
Walmart plastic bag	76.78	1.10
PLA/starch/lecithin film	59.250	1.01

**Table 4 polymers-17-02488-t004:** Comparison of physico-mechanical properties of WBAX bioplastic and LDPE.

Property	WBAX Bioplastic	LDPE (Synthetic Plastic)
Tensile Strength (MPa)	3.34	10–30
Elongation at Break (%)	138	100–600
Water Contact Angle (°)	80	>95
Water Solubility (%)	~40	<1
Biodegradability	Yes	No
Acid Resistance	Low concentration	Moderate
Alkali Resistance	High concentration	Moderate
Transparency	Yes	Yes

## Data Availability

The original contributions presented in this study are included in the article. Further inquiries can be directed to the corresponding author.

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
