# Peer review of "Arabinoxylan-Based Bioplastic from Wheat Bran: A Promising Replacement for Synthetic Plastics"

_polymers, 2025, doi:10.3390/polym17182488_

Round 1
Reviewer 1 Report
Comments and Suggestions for Authors
Dear Author(s),
I have completed evaluation of the manuscript submitted to Polymers, entitled “Arabinoxylan-Based Bioplastic from Wheat Bran: A Promising Replacement for Synthetic Plastics” (Manuscript Number: Polymers 2024, 16, x).
The manuscript reports on the development and characterization of a biodegradable polymeric material derived from wheat bran, with particular emphasis on its prospective utility in food packaging applications. The study predominantly focuses on the extraction of arabinoxylan from wheat bran and its subsequent integration into a biopolymeric matrix via biosynthetic processes.
This work holds notable relevance to the broader field of biopolymer and green composite research, particularly among those engaged in the synthesis, application development, and physicochemical characterization of sustainable materials. The research presents original contributions and offers valuable insights into the utilization of agro-industrial waste for bioplastic production.
However, prior to considering the manuscript suitable for publication, I recommend addressing the following scientific and structural concerns:
- Mechanical characterization: It would be beneficial to include tensile strength, tear resistance, and strain-at-break (extensibility/stretchability) data to substantiate the material's suitability for packaging applications.
- Tactile evaluation: Consider incorporating tactile response or haptic property analysis, as it would provide meaningful data regarding the handling characteristics of the biopolymer, especially in consumer-oriented applications.
- Process visualization: A schematic diagram illustrating the processing workflow or fabrication route of the bioplastic material would enhance clarity and reproducibility.
- Graphical quality: Please improve the resolution and visual clarity of spectral and analytical data figures (e.g., FTIR, SEM, or others), to facilitate more accurate interpretation of the results.
In conclusion, the manuscript addresses a timely and significant topic within the domain of sustainable biomaterials and contributes to the advancement of bioplastic technologies for potential implementation in the food packaging sector. With incorporation of the aforementioned revisions, the manuscript may be deemed suitable for publication in Polymers.
Sincerely,
Author Response
Reviewer-1
I have completed the evaluation of the manuscript submitted to Polymers, entitled “Arabinoxylan-Based Bioplastic from Wheat Bran: A Promising Replacement for Synthetic Plastics” (Manuscript Number: Polymers 2024, 16, x).
The manuscript reports on the development and characterization of a biodegradable polymeric material derived from wheat bran, with particular emphasis on its prospective utility in food packaging applications. The study predominantly focuses on the extraction of arabinoxylan from wheat bran and its subsequent integration into a biopolymeric matrix via biosynthetic processes.
This work holds notable relevance to the broader field of biopolymer and green composite research, particularly among those engaged in the synthesis, application development, and physicochemical characterization of sustainable materials. The research presents original contributions and offers valuable insights into the utilization of agro-industrial waste for bioplastic production.
However, prior to considering the manuscript suitable for publication, I recommend addressing the following scientific and structural concerns:
- Mechanical characterization: It would be beneficial to include tensile strength, tear resistance, and strain-at-break (extensibility/stretchability) data to substantiate the material's suitability for packaging applications.
Thank you for your concern. The tensile strength values have now been included in the graph. Although we did not specifically evaluate tear resistance, we have discussed its relevance in comparison with the other measured mechanical properties.
- Tactile evaluation: Consider incorporating tactile response or haptic property analysis, as it would provide meaningful data regarding the handling characteristics of the biopolymer, especially in consumer-oriented applications.
We appreciate your valuable suggestion regarding tactile response and haptic property analysis. While such an evaluation would indeed provide meaningful insights into consumer handling characteristics, our current study focused primarily on fundamental mechanical properties such as tensile strength, elongation at break, flexibility, and optical transparency, which already provide an indirect indication of handling performance.
- Process visualization: A schematic diagram illustrating the processing workflow or fabrication route of the bioplastic material would enhance clarity and reproducibility.
A schematic diagram with details experimental process for arabinoxylan and bioplastic synthesis has been included in the manuscript.
- Graphical quality: Please improve the resolution and visual clarity of spectral and analytical data figures (e.g., FTIR, SEM, or others), to facilitate more accurate interpretation of the results.
Image qualities have been improved.
In conclusion, the manuscript addresses a timely and significant topic within the domain of sustainable biomaterials and contributes to the advancement of bioplastic technologies for potential implementation in the food packaging sector. With incorporation of the aforementioned revisions, the manuscript may be deemed suitable for publication in Polymers.
Reviewer 2 Report
Comments and Suggestions for Authors
Overall, the study looks relevant and suitable for the journal’s scope.
However, my primary comment is that the presented material’s scientific substantiation is not enough. It looks like a technical report on bioplastic producing for specific engineering applications.
A research paper must contain novel scientific knowledge, an exact explanation of why the new materials demonstrate observed properties. For example, it can be implemented via more detailed discussion of FTIR spectroscopy results and surface exploration by microphotographs.
Section "4. Discussion" should be expanded. It is necessary to describe why different composites interact with alkalis and acids differently, according to data of the change in the chemical composition of different composites. It is necessary to explain why the hydrophilicity of the resulting material changes.
Thus, after suggested work is done, it will be necessary to formulate new purpose – not just obtaining new materials, but obtaining bioplastics with specified properties. And the conclusions section will also require revision.
Minor comments.
Line 88. It would be useful to describe the method of Arabinoxylan extraction briefly.
Section "2.2.2. Bioplastic Synthesis". Is it possible to provide chemical formulas for the transformations of substances in obtaining the new material? However, it is better to do this at line 234, for the description of "interactions between arabinoxylan, polyvinyl alcohol, and glycerol". These formulas can answer the question of why different composites have different properties in their interaction with acids and alkalis.
Lines 171 - 172. The choice of "A Ziploc bag and a polyethylene bag" as comparison samples is not obvious. Why is ordinary paper not also used?
Author Response
Reviewer -2
Overall, the study looks relevant and suitable for the journal’s scope.
However, my primary comment is that the presented material’s scientific substantiation is not enough. It looks like a technical report on bioplastic producing for specific engineering applications.
A research paper must contain novel scientific knowledge, an exact explanation of why the new materials demonstrate observed properties. For example, it can be implemented via more detailed discussion of FTIR spectroscopy results and surface exploration by microphotographs.
Thank you for your comment. The discussion section has been added in detail, describing the reasoning.
Section "4. Discussion" should be expanded. It is necessary to describe why different composites interact with alkalis and acids differently, according to data of the change in the chemical composition of different composites. It is necessary to explain why the hydrophilicity of the resulting material changes.
The discussion section has been revised, addressing the reasoning
Thus, after suggested work is done, it will be necessary to formulate new purpose – not just obtaining new materials, but obtaining bioplastics with specified properties. And the conclusions section will also require revision.
The conclusion section has been modified according to your comments.
Minor comments.
Line 88. It would be useful to describe the method of Arabinoxylan extraction briefly.
Detailed experimental conditions and process have been explained with a schematic diagram.
Section "2.2.2. Bioplastic Synthesis". Is it possible to provide chemical formulas for the transformations of substances in obtaining the new material? However, it is better to do this at line 234, for the description of "interactions between arabinoxylan, polyvinyl alcohol, and glycerol". These formulas can answer the question of why different composites have different properties in their interaction with acids and alkalis.
Chemical structure and the interaction among the blends have been addressed with the figure.
Lines 171 - 172. The choice of "A Ziploc bag and a polyethylene bag" as comparison samples is not obvious. Why is ordinary paper not also used?
Thank you for your kind comment. We chose Ziploc and polyethylene bags as they are common commercial packaging benchmarks. Paper was not included since its mechanical and barrier properties differ greatly from polymer-based films
Reviewer 3 Report
Comments and Suggestions for Authors
The manuscript reports on arabinoxylan-based bioplastics from wheat bran. This is an interesting subject in development of biopolymers. The manuscript is written well and easy to follow. However, some detailed comments as below need to be addressed:
1)Abstract: Some important results in the form of values should be presented in the Abstract.
2)L29-34: Introduction-Please include some examples for each category of the bioplastics.
3)L51: Please double check this number (90%) with more references: ‘…90% of the wheat bran remains in landfills,…’
4)L91-92: How arabinoxylan was extracted? All details and parameters need to be presented.
5)L93: What was the ratio of arabinoxylan and water? Mass or volumes?
6)L84: How much glycerol was added?
7)L95-96: What was the ratio?
8)L101: What equipment was used? How much pressure was applied?
9)L192: What are these time intervals (days)?
10)L210: How samples were dried? What condition?
11)Why no experimental design/statistical analysis was used in this study?
12)A one-way ANOVA and a multiple-range ranking test (such as Tuckey)would be helpful to analyze the dependent variables such as tensile strength.
13)Fig 2: Error bars (for standard deviation-SD) should be presented on the columns.
14)Table 3: Please include SD values for each mean value that already presented.
15)Fig 5 and 6: SD values should be presented on the columns.
16(More references required under Results and Discussion to compare the results of this study with.
Author Response
Reviewer -3
The manuscript reports on arabinoxylan-based bioplastics from wheat bran. This is an interesting subject in development of biopolymers. The manuscript is written well and easy to follow. However, some detailed comments as below need to be addressed:
1)Abstract: Some important results in the form of values should be presented in the Abstract.
Thank you for your kind observation. It has been revised.
2)L29-34: Introduction-Please include some examples for each category of the bioplastics.
An example for each categoric of plastic has been included
3)L51: Please double check this number (90%) with more references: ‘…90% of the wheat bran remains in landfills,…’
This paragraph has been revised with references.
4)L91-92: How arabinoxylan was extracted? All details and parameters need to be presented.
A schematic diagram has been added.
5)L93: What was the ratio of arabinoxylan and water? Mass or volumes?
It has been revised.
6)L84: How much glycerol was added?
Ratio of glycerol, PVA, and arabinoxylan has been described in Table 1. The water arabinoxylan ratio has been mentioned in the experimental section.
7)L95-96: What was the ratio?
Ratios are given in Table 1
8)L101: What equipment was used? How much pressure was applied?
It has been revised.
9)L192: What are these time intervals (days)?
Time interval has been added now.
10)L210: How samples were dried? What condition?
The condition has been mentioned.
11)Why no experimental design/statistical analysis was used in this study?
Thank you for the comment. As this was an exploratory study focused on synthesis and characterization, no experimental design or advanced statistical analysis was applied. Replicates were used to ensure reliability
12)A one-way ANOVA and a multiple-range ranking test (such as Tuckey)would be helpful to analyze the dependent variables such as tensile strength.
We appreciate the suggestion. We agree that ANOVA and Tukey’s test would strengthen the analysis, while this study focused on baseline characterization.
13)Fig 2: Error bars (for standard deviation-SD) should be presented on the columns.
SD has been included in the bar chart
14)Table 3: Please include SD values for each mean value that already presented.
SD value has been included.
15)Fig 5 and 6: SD values should be presented on the columns.
SD has been included in the bar chart
16(More references required under Results and Discussion to compare the results of this study with.
Thank you for your concern. We have added additional references in the relevant sections to ensure better comparability with our study.
Round 2
Reviewer 2 Report
Comments and Suggestions for Authors
The authors have significantly altered their manuscript. I saw a typo in line 283. The name of the figure is not 1b, it is 4b.
Author Response
The authors have significantly altered their manuscript. I saw a typo in line 283. The name of the figure is not 1b, it is 4b.
Thank you so much for carefully checking the manuscript and for pointing out this error. The typo in line 283 has been corrected; the figure reference now reads Figure 4b instead of Figure 1b in the revised version.
Reviewer 3 Report
Comments and Suggestions for Authors
It is not easy to track the revisions in the new version. All new revisions need to be indicated that correspond to which lines and pages in the new version in the replies.
Author Response
It is not easy to track the revisions in the new version. All new revisions need to be indicated that correspond to which lines and pages in the new version in the replies.
Thank you for your observation. We apologize for the difficulty in tracking the revisions. Some sections have been revised as suggested, which has altered the line and page numbers in the new version. To address this, we have prepared a table that indicates the original comments with their corresponding line and page numbers, as well as the updated page and line numbers after revision. We hope this will make it easier to track the changes.
Comment |
Page |
Line |
Response |
Page |
Line |
1)Abstract: Some important results in the form of values should be presented in the Abstract. |
P1 |
-- |
Thank you for your kind observation. It has been revised. |
P1 |
L14,15 |
2)L29-34: Introduction-Please include some examples for each category of the bioplastics.
|
P1 |
L29-34: |
Example for each categoric plastic has been included and the sentence has been revised.
|
P1 |
L29-39 |
3)L51: Please double check this number (90%) with more references: ‘…90% of the wheat bran remains in landfills,…’
|
P2 |
L51 |
This paragraph has been revised with references.
|
P2 |
L54-57 |
4)L91-92: How arabinoxylan was extracted? All details and parameters need to be presented.
|
P2 |
L91-92 |
A schematic diagram has been added.
|
P2,3 |
L93-105 |
5)L93: What was the ratio of arabinoxylan and water? Mass or volumes?
|
P2 |
L93 |
It has been revised.
|
P3 |
L113 |
6)L84: How much glycerol was added?
|
P2 |
L84 |
Ratio of glycerol, PVA, arabinoxylan has been described in table 1. The water, arabinoxylan ratio has been mentioned in the experimental section.
|
P4 |
Table 1 |
7)L95-96: What was the ratio?
|
P2 |
L95-96 |
Ratios are given in table 1
|
P4 |
Table 1 |
8)L101: What equipment was used? How much pressure was applied?
|
P3 |
L101 |
It has been revised.
|
P3 |
L121,122 |
9)L192: What are these time intervals (days)?
|
P5 |
L192 |
Time interval has been added now.
|
P7 |
L250-253 |
10)L210: How samples were dried? What condition?
|
P5 |
L210 |
Condition has been mentioned.
|
P7 |
L268 |
11)Why no experimental design/statistical analysis was used in this study?
|
|
|
Thank you for the comment. As this was an exploratory study focused on synthesis and characterization, no experimental design or advanced statistical analysis was applied. Replicates were used to ensure reliability
|
|
|
12)A one-way ANOVA and a multiple-range ranking test (such as Tuckey)would be helpful to analyze the dependent variables such as tensile strength.
|
|
|
We appreciate the suggestion. We agree that ANOVA and Tukey’s test would strengthen the analysis, while this study focused on baseline characterization.
|
|
|
13)Fig 2: Error bars (for standard deviation-SD) should be presented on the columns.
|
P7 |
Fig 2 |
SD , has been included in the bar chart
|
P10 |
FIG 5 |
14)Table 3: Please include SD values for each mean value that already presented.
|
P8 |
Table 3 |
SD value has been included.
|
P12 |
Table 3 |
15)Fig 5 and 6: SD values should be presented on the columns.
|
P10,11 |
Fig 5,6 |
SD , has been included in the bar chart
|
P14,15 |
FIG 8,9 |
16(More references required under Results and Discussion to compare the results of this study with.
|
P14,15 |
L359-392; L396-408 |
Thank you for your concern. We have added additional references in the relevant sections to ensure better comparability with our study.
|
P18-20 |
L534-590; P594-606 |

Round 3
Reviewer 3 Report
Comments and Suggestions for Authors
The comments are addressed carefully. The manuscript is in good shape now and it is suggested for publication.